# The Discharge Forecasting of Multiple Monitoring Station for Humber River by Hybrid LSTM Models

**Yue Zhang** [1], **Zhaohui Gu** [1], **Jesse Van Griensven Thé** [2], **Simon X. Yang** [1,*] and **Bahram Gharabaghi** [1,*]

1   School of Engineering, University of Guelph, 50 Stone Road East, Guelph, ON N1G 2W1, Canada;
    zhang26@uoguelph.ca (Y.Z.); zgu04@uoguelph.ca (Z.G.)
2   Lakes Environmental, 170 Columbia St. W, Waterloo, ON N2L 3L3, Canada; jesse.the@weblakes.com
*   Correspondence: syang@uoguelph.ca (S.X.Y.); bgharaba@uoguelph.ca (B.G.)

**Abstract:** An early warning flood forecasting system that uses machine-learning models can be utilized for saving lives from floods, which are now exacerbated due to climate change. Flood forecasting is carried out by determining the river discharge and water level using hydrologic models at the target sites. If the water level and discharge are forecasted to reach dangerous levels, the flood forecasting system sends warning messages to residents in flood-prone areas. In the past, hybrid Long Short-Term Memory (LSTM) models have been successfully used for the time series forecasting. However, the prediction errors grow exponentially with the forecasting period, making the forecast unreliable as an early warning tool with enough lead time. Therefore, this research aimed to improve the accuracy of flood forecasting models by employing real-time monitoring network datasets and establishing temporal and spatial links between adjacent monitoring stations. We evaluated the performance of the LSTM, the Convolutional Neural Networks LSTM (CNN-LSTM), the Convolutional LSTM (ConvLSTM), and the Spatio-Temporal Attention LSTM (STA-LSTM) models for flood forecasting. The dataset, employed for validation, includes hourly discharge records, from 2012 to 2017, on six stations of the Humber River in the City of Toronto, Canada. Experiments included forecasting for both 6 and 12 h ahead, using discharge data as input for the past 24 h. The STA-LSTM model's performance was superior to the CNN-LSTM, the ConvLSTM, and the basic LSTM models when the forecast time was longer than 6 h.

**Keywords:** flood forecasting; LSTM; CNN-LSTM; ConvLSTM; STA-LSTM; discharge; water level; spatio-temporal series

## 1. Introduction

Historically, floods in Canada occur in the spring, due to snowmelt, and in the summer, due to intense thunderstorms [1–3]. Urbanization has amplified the flood risks due to rapid runoff from impervious surfaces. Floods jeopardize lives, and inundation-prone areas suffer devastating economic losses. In this context, governments rely on early flood warning and forecasting systems to help protect lives and prevent property damage by deploying countermeasures [4]. The flow chart in Figure 1 illustrates a typical flood warning system.

Compared to traditional solutions, which assess flood risks, early flood forecasting and warning systems play a more significant role in alerting people of imminent floods [5,6]. Early flood warning systems usually come with different lead times, which play a critical factor in the control and mitigation of risks during a flood hazard and related disasters. Such multi-functional forecasting systems enhance community preparedness in the context of floods and minimize losses that usually follow a flood. The system will typically predict the scale, timing, location, and likely damages of the flood [7]. It draws data all year round from sensors placed in strategic points of the water basins, such as in lakes and rivers, or on flood defenses such as dams, dikes, embankments, or specially constructed structures for flood forecasting and monitoring. Promising preventive measures require extensive

collaboration across multiple disciplines, such as deep learning algorithms, remote sensing, hydrology, and meteorology [8,9]. Forecasting model integrity is enhanced due to the collaboration of such disciplines. These forecast models are developed and managed by assessing flood risks, local hazard monitoring, flood risk dissemination services, and community response [10].

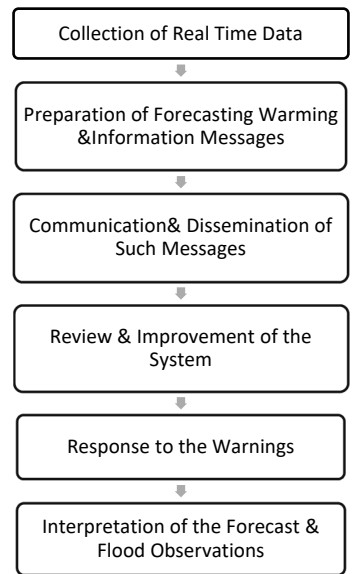

**Figure 1.** Components of the Flood Forecasting and Warning System.

It is common for countries to deploy large-scale sensor networks, given the flood dis-asters previously faced [11]. Large-scale sensor networks collect critical data on water bodies, such as water velocity, temperature flooding, etc. The growing availability of such data, combined with the need to prepare for flood situations, has pushed researchers to analyze how existing computational resources can improve forecast accuracy. The popularity of deep learning and machine learning technologies enables the transformation of practical knowledge into actionable ones. "Emerging advances in computing technologies, coupled with big-data mining, have boosted data-driven applications, among which Machine Learning (ML) technologies bearing flexibility and scalability in pattern extraction has modernized scientific thinking and predictive applications" [12]. Technologies with deep learning, big-data mining, aggregation, and model ensemble drive methodologically oriented countermeasures that aid in forecasting certain hydrological parameters. The parameters included are reservoir inflow, reservoir inflow, river flow parameters, tropical cyclone tracking, and anticipating different lead times in flooding.

Deep learning is a subset of machine learning, which employs multiple layers of neural networks that can gain knowledge and acquire skills akin to the working of the human brain [13,14]. The proliferation of deep learning algorithms has given rise to an improvement in deep learning capabilities. The popularity of deep learning, for forecasting, stems from the fact that data, in the real world, evolve with time and are represented as time-series problems [15]. These are highly diverse, unstructured, inter-connected, and contain spatio-temporal patterns. Deep neural networks can handle complex time series better, offering robust computational facilities in the form of advanced data processing, reduced complexity in data processing, and improved accuracy in model prediction [16].

Physical hydrological models, such as the Environmental Protection Agency Storm Water Management Model (EPA-SWMM) were developed to simulate flooding events in cities. Still, these models are not available for older and larger megacities due to the SWMM relay to the high precision mapping and a simulation of the underground drainage system [17]. Then, the semi-physical hydrological models, such as the Cellular Automata (CA), need correlation calculation and an abundance of analysis for the datasets [18]. This

way, researchers combine the rainfall-runoff model and a flood-level map database to improve the alert system. They can momentarily estimate the spatial distribution of flood depth by engaging GIS tools in the flood area [19–21]. There are no solutions for the spatial depth measurement of rainstorm events. Therefore, our research focus is on improving the performance of hybrid Long Short-Term Memory (LSTM) hydrological models.

Missing values or corrupt values typically cause a lack of uniformity in a normal situation. Still, in the case of flood forecasting, the lack of uniformity is caused by sporadically available data and at either increasing or decreasing time-space intervals [22]. This makes prediction accuracy a problem in flood forecasting. Such sporadic data with time-series problems are also present in many forecasting and prediction areas. The deep neural network contains the required structure to control the complexity, especially CNN. Many scientific domains benefit from a wealth of satellite and model output data because huge amount of data are needed to fit [23]. Spatio-temporal data and time-series problems are huge challenges for space technology developments [16,24,25]. Spatio-temporal data, characterized by complex information and heterogeneous aspects, can create uncertainties where a network topology might not scale [26,27].

The spatio-temporal nature of data is perhaps the biggest challenge when it comes to flood forecasting with CNN and LSTM. Spatio-temporal data has three dimensions: the two-dimensional spatial and the temporal dimension [28]. In the quantitative analysis end, collected time-series data may be used to capture geographical processes, such as in flood forecasting systems, at some defined regular interval [29,30]. It might also happen at irregular intervals, such as in the case of continuous daily occurrences or discrete occurrences, as to when an event might occur randomly in the temporal scale [31].

Spatio-temporal data is not that easy to access, nor is it smooth. It has both local correlations, as well as gradients. In addition, there are spatio-temporal mutations. "As the accumulation of spatio-temporal data, the low-quality problems of multivariate spatio-temporal data become clear and mainly present numerous missing data, high noise of time series and great different spatial scale of spatiotemporal data" [29,30]. Therefore, spatio-temporal data preprocessing can help improve prediction accuracy.

Artificial Neural Network (ANN) refers to a complex network structure formed by many interconnected processing units (neurons) [32,33]. A form of ANN called the Recurrent Neural Network (RNN) is used frequently in forecasting. RNN has arbitrary connections between the neurons, and the recurrent connections allow memory to be persisted in the internal state. Unlike an ANN, the independent layers in RNN are converted to the dependent layer [34]. The same bias and weight are used. All hidden layers are joined up as a single recurrent layer. This enables the RNN to process inputs of any given length. As more layers are added using specific activation functions, the gradients of the loss function approach zero, which makes the network hard to train [35]. This is inevitable, as some activation functions, such as the sigmoid, for instance, will manage large input spaces in smaller input spaces, and this will change the output. In turn, the derivative becomes smaller and, over time, will exponentially decay. The learning of long-term data sequences is, therefore, hampered. As the gradient that carries information in RNN becomes smaller and the parameter updates to new inputs become negligible, there is no real learning. Therefore, the forecasting benefits of RNN are hindered. A solution to this vanishing gradient problem in RNN is the LSTM networks [36].

LSTM is an improvement over simple RNN, which captures sequential data long-term dependence [37–39]. The LSTM architecture includes specially designed gates, units, and memory cells [40–42] that learn and retain the state of information, while deciding when to forget irrelevant data [43]. Simple RNNs, in comparison, only update a single past state. The cell state, serving as the system's memory, restores the system's capability through backpropagation and time algorithms [44,45].

LSTM models employ these neuron gates to learn, forget, and surround cell memory for better control of information flow. With training, the input gate becomes proficient as a control for the input that must be remembered for a certain period [45].

Advanced deep learning methods, such as Convolutional Neural Network (CNN) and LSTM, offer a better spatio-temporal series prediction than a simple time series prediction, for the extraction of abstract and high-level information, from images and complex data [46]. ConvLSTM is a variant of LSTM in that it has the convolution operation inside the LSTM cell. On the other hand, both models are special types of RNN and have deep learning capabilities [47,48]. ConvLSTM replaces the matrix multiplication operation at the gates, to better capture underlying spatial features with convolution, and varies from LSTM based on the input dimensions. It is specifically designed for 3D data. CNNLSTM is an integration of CNN with LSTM. As such, the CNN model is specifically integrated with LSTM, allowing the CNN model to process data and then use LSTM for processing the one-dimensional result feed, since LSTM cannot process multiple dimensions.

*Contributions*

The spatio-temporal attention LSTM model combines the LSTM structure and spatio-temporal attention module to selectively use the critical and useful hydrological features [43]. For the spatio-temporal attention LSTM model (STA-LSTM), the main LSTM network is used for feature extraction, temporal correlation, and final classification. The temporal attention is used to assign appropriate importance to different frames, and the spatial attention is used to assign appropriate priority to other nodes [49].

Almost all researchers' hydrological predictions employ a single monitoring station, producing a time-series prediction. However, rivers contain many monitoring stations, where adjacent monitoring stations' discharge values correlate. Flood forecasting is a spatio-temporal prediction problem. One should build the relationship between the upstream monitoring stations and the downstream monitoring stations that is important to improve the accuracy of flood prediction when an earlier warning is needed. Urban flood prediction is the beginning of flood forecasting. We find the most important factors to be the discharge of upstream monitoring stations and precipitation for the hybrid models [50]. Due to complex spatio-temporal datasets and high accuracy, the system needs more efficient models.

Therefore, our research aims to expand the time series prediction to include spatial information to improve forecasting. To achieve this objective, this research adopted modified hybrid LSTM variations [51], such as Convolutional Neural Networks LSTM model (CNN-LSTM), Convolutional LSTM model (ConvLSTM), and Spatio-temporal Attention LSTM model (STA-LSTM) [43].

## 2. Materials and Methods

### 2.1. Study Area and Materials

The Humber River is one of the most important rivers in Southern Ontario, Canada. It is a tributary of Lake Ontario, and it is one of two major rivers on either side of the city of Toronto. The flood forecasting of the Humber River greatly influences the western parts of metropolitan Toronto. Humber River's Station 02HC003 is the nearest to Toronto, making it a key piece in the flood forecasting for this metropolitan area.

The Humber River flows right through downtown Toronto, so the discharge prediction of the south of Humber River will be crucial to protecting human life and property. The network of real-time tipping-bucket rainfall monitoring in the Humber River watershed is sparse and does not accurately capture the spatial variability of the intense, localized summer thunderstorms. Therefore, the raw rainfall data must, first, be pre-processed and accumulated for the watershed, over time and space, before it can provide meaningful input to the machine learning model. Therefore, to keep the flood forecasting system simple and practical, yet fairly accurate, we decided not to include rainfall monitoring data as part of the scope of this manuscript.

Moreover, the perfect early flood forecasting system could better coordinate the interests of the government, the affected people, and the insurance industry in the sharing of

flood losses. Due to the trend of insured catastrophic losses increasing year by year, the high accuracy early flood forecasting system should be accessible to everyone immediately.

From Figure 2, we can identify that Stations 02HC025, 02HC031, 02HC032, and 02HC047 are in the headwaters of the Humber River Watershed and upstream of the critical station 02HC003 which is located in the flood-prone areas of downtown Toronto near the mouth of the Humber River watershed.

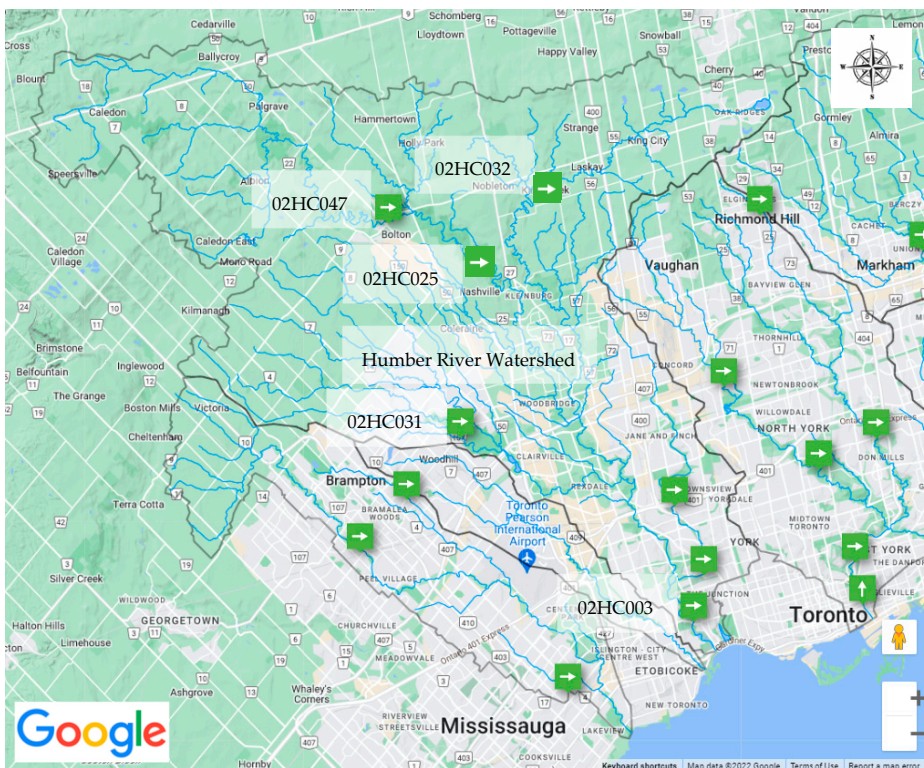

**Figure 2.** The network of real-time hydrometric monitoring stations in the Humber River Watershed. (Source: http://beta.trcagauging.ca/ accessed on 20 April 2022).

The hydrological prediction models perform well on the time series data, so they were evaluated for the spatio-temporal data. Since the STA-LSTM has a good performance on the flood prediction with time-series data, we will compare the performance of the CNN-LSTM model, ConvLSTM model, and STA-LSTM with the spatio-temporal data for flood forecasting. We applied LSTM based models because they are highly capable of dealing with spatio-temporal series-based data sequences as compared to traditional models (e.g., M5MT, extreme learning machine (ELM), SVM) [52]. These models are based on automatic feature learning and consider the previous information during training.

Due to the level of urbanization and size of the Humber River watershed, the catchment response time typically ranges from 5 to 10 h, depending on the rainfall storm event type and duration (e.g., short-burst but intense summer thunder storms versus longer duration rainfall combined with snowmelt events in the spring). The Flood warnings for the Humber River watershed must be issued to a range of users and for various purposes. These purposes may include: readying operational teams and emergency personnel, warning the public of the timing and location of the event, and in extreme cases, to enable preparation for undertaking evacuation and emergency procedures. Therefore, we train/test the model for 12-h-ahead and 24-h-ahead forecast scenarios and evaluate model accuracy.

*2.2. Datasets and Data Preprocessing*

Six years of hourly dataset from five stations will be used, provided by stations 02HC047, 02HC032, 02HC031, 02HC025, and 02HC003. The five stations are located in the west of Toronto. We would predict the discharge (unit: m$^3$/s) of station 02HC003, which is in the flood-prone areas of Toronto, and we will observe the mean square error (*MSE*) and mean absolute error (*MAE*) to evaluate the forecasting performance.

We selected 70 percent of the dataset for training, covering the period from 2012 to 2014. The following 20 percent of the data is used for testing, covering the period from 2016 to 2017. The remaining 10 percent of the data is reserved for validation, covering the 2015 year. We used five stations to test the four kinds of hybrid LSTM models and compare their mean square error. We evaluated forecasts from 1-h ahead to 12-h ahead, employing the past 24-h monitoring as input.

*2.3. Models*

The LSTM, ConvLSTM, and CNN-LSTM models were implemented in TensorFlow, using Keras library in Python, and the STA-LSTM was implemented in torch.nn by using the nn.Module in Python. A batch size of 50 and 200 epochs has been used in the research because the optimal epochs could prevent the model from overfitting or underfitting.

1. A LSTM is a neural network that accounts for dependencies in a spatio-temporal series, which is commonly used for forecasting purposes. The altering of flood forecasting is from a time series prediction to a spatio-temporal series prediction. The LSTM model is a good choice for the beginning. The structure of the LSTM model is presented in Figure 3, which is comprised of LSTM layers. In the structure, input sequences are provided to the input layer followed by two LSTM layers. The dropout layer is added to prevent the model from overfitting, and then, two LSTM layers are added, followed by a Flatten layer. Three dense layers are added, followed by one dropout and three dense layers, which are used to change the dimensions of the vectors. Finally, the last output layer returns the output sequences. The equations of each layer of the LSTM model are given as:

$$i_t = \sigma(W_i \cdot [h_{t-1}, x_t] + W_{ci} \cdot C_{t-1} + b_i), \tag{1}$$

$$o_t = \sigma(W_o \cdot [h_{t-1}, x_t] + W_{co} \cdot C_{t-1} + b_t), \tag{2}$$

$$f_t = \sigma\left(W_f \cdot [h_{t-1}, x_t] + W_{cf} \cdot C_{t-1} + b_f\right), \tag{3}$$

$$\widetilde{C}_t = \tanh(W_C h_{t-1} + W_C x_t + b_C), \tag{4}$$

$$C_t = f_t C_{t-1} + i_t \widetilde{C}_t, \tag{5}$$

$$h_t = o_t \tanh(C_t), \tag{6}$$

   where $i_t$ represents the input gate, $o_t$ represents the output gate, $f_t$ represents the forget gate, $\widetilde{C}_t$ represents the memory cell, $h_t$ represents a hidden state, $h_{t-1}$ and $C_{t-1}$ represent the inputs of previous timestamps, $x_t$ denotes the current timestamp, $\sigma$ symbolizes the sigmoid function, $b_x$ represents the bias of respective gates, and $W_x$ represents the weights of respective gates.

2. ConvLSTM is a repetitive neural network for spatial-temporal prediction with state-of-the-art, state-to-state, and phase-to-phase convolutional characteristics. Figure 4 shows the structure of a ConvLSTM model. ConvLSTM predicts the future state of a grid cell based on the income and historical status of its neighbors [53,54]. The ConvLSTM can keep the input features as three-dimensional (3D), and it still reserves the advantages of Fully Connected-LSTM [55]. In this study, we used a ConvLSTM model with two convolutional layers and LSTM layers. After providing the sequences of the input layer, two convolutional layers are added, followed by one dropout layer, to avoid the overfitting. Preceding this, two LSTM layers are added to make the

ConvLSTM model followed by a flatten layer. Six dense layers are added to the model, and a dropout layer is added in the middle. The output layer provided the output sequences [55].

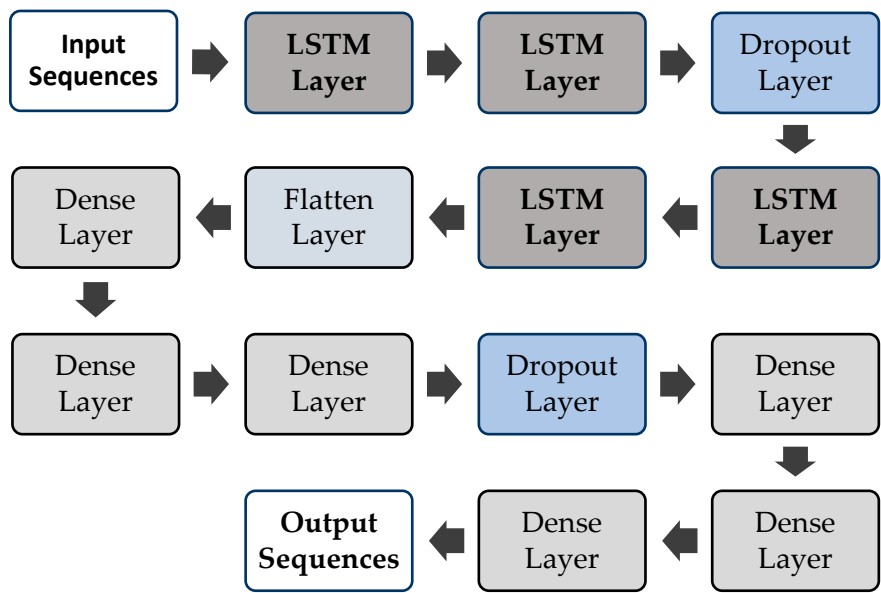

**Figure 3.** LSTM Model Structure [53].

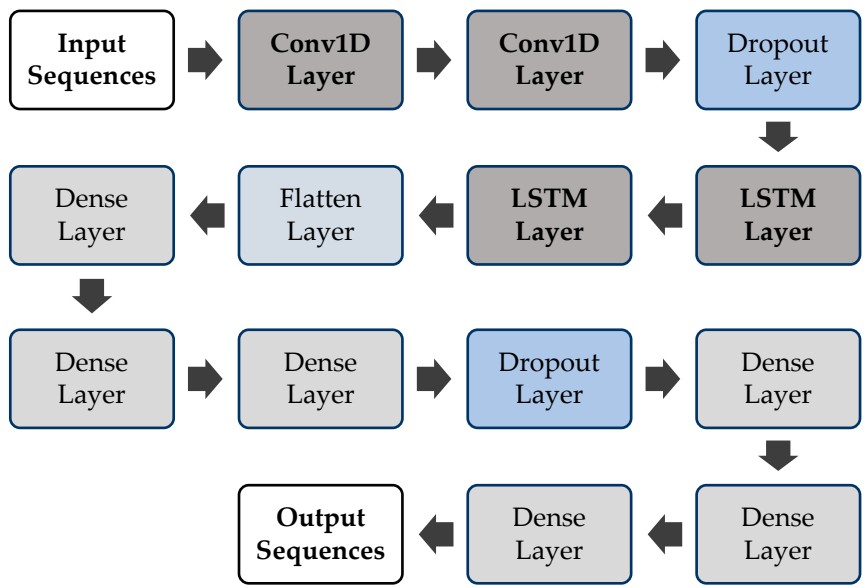

**Figure 4.** ConvLSTM Model Structure [53].

3.  CNN-LSTM was initially known as a Long-term Recurrent Convolutional Network (LRCN) model, but in this article, we will use the most common term, "CNN-LSTM", to refer to LSTMs that employs a CNN as a front end [56]. The LSTM model can process the dataset of CNN and the LSTM sequences that come from the one-dimensional result of CNN. The structure of a CNN-LSTM model used in this study is shown in Figure 5. We used four CNN-LSTM layers with a combination of other layers, including dropout, flatten, and dense layers. Two ConvLSTM2D layers are added followed by a dropout layer, and then, two more ConvLSTM2D layers are included. The remaining setup of layers is similar to the previously discussed models.

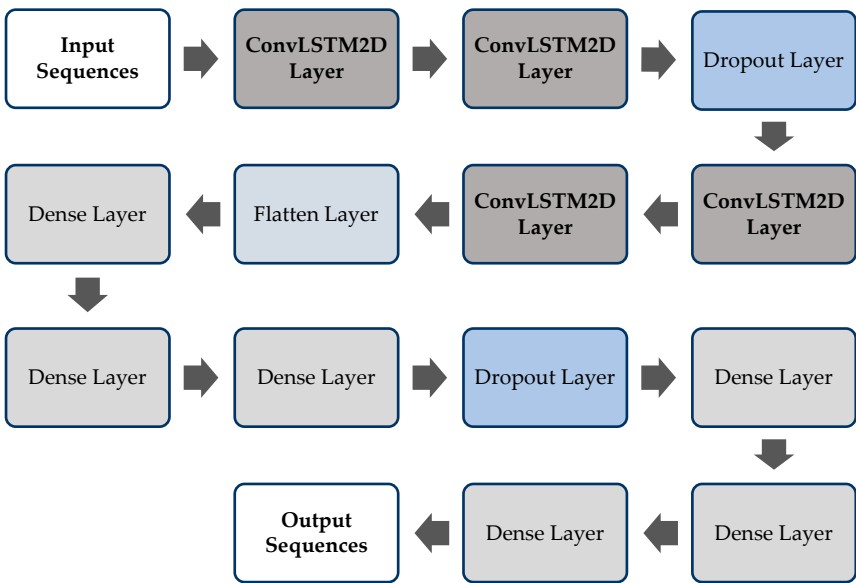

**Figure 5.** CNN-LSTM Model Structure [53].

4. The STA-LSTM is the spatial attention operation and the temporal attention operation introduced into the LSTM cell to make full use of the spatio-temporal information of the input. The spatial attention operation works for the input features and the temporal attention operation works with the hidden layer of the LSTM. Therefore, the spatial attention weights and the temporal attention weights affect the inputs and the output, respectively [43]. Debugging the Spatial attention weights and the temporal attention weights is the main method to improve the performance of the STA-LSTM model. Figure 6 shows the structure of an STA-LSTM model developed in this study. The input sequences are provided to the spatial attention module, which is comprised of three layers, including linear, sigmoid, and softmax layers. After that, the LSTM layer is added, followed by the hidden layer, and passed the information to the temporal attention module, which consists of linear, ReLu, and softmax activation functions.

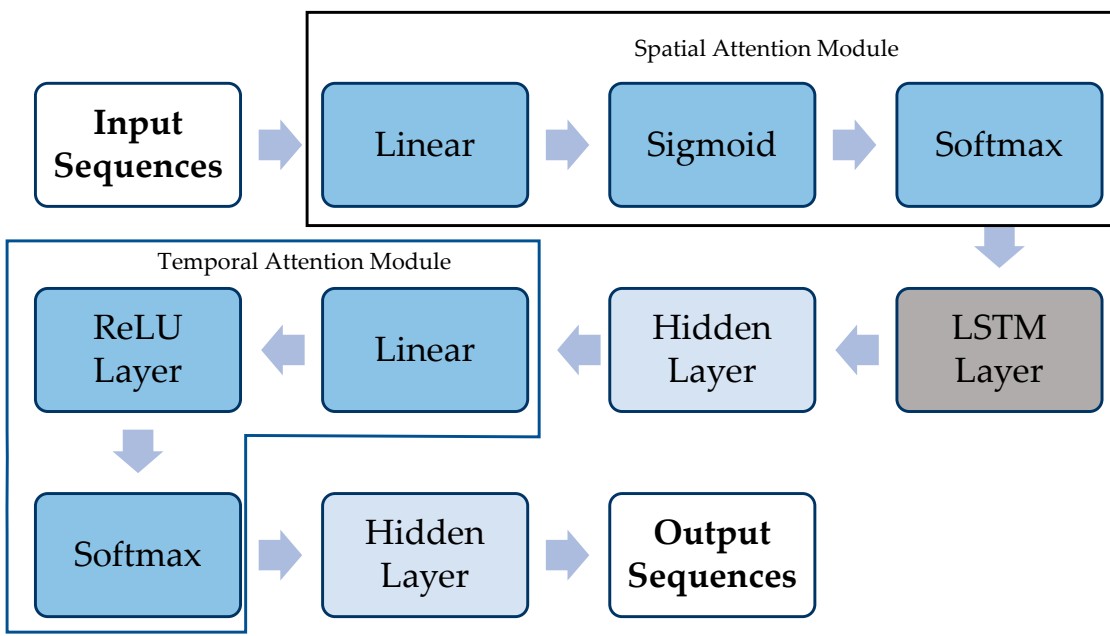

**Figure 6.** STA-LSTM Model Structure [43].

*2.4. Evaluation Measures*

Three evaluation measures are used to evaluate the performance of proposed models. Each evaluation functions are described below:

1.  The Mean Square Error function is defined as:

$$MSE = \frac{1}{n} \sum_{i=1}^{n} (Y_i - \hat{Y}_i)^2, \tag{7}$$

where $n$ is the total number of data points, $Y_i$ is the observed value, and $\hat{Y}_i$ is the predicate value.

2.  We also use Mean Absolute Error ($MAE$), defined in (8), to evaluate the purposed model's performance because there are some outliers for each station that measures flood season. The meaning of the same symbols is the same as that in the $MSE$. It is well known that the median is more robust than the mean for outliers, so $MAE$ is more stable for outliers than $MSE$.

$$MAE = \frac{1}{n} \sum_{i=1}^{n} |Y_i - \hat{Y}_i|. \tag{8}$$

3.  The *Error Rate* is used as the third evaluation measure concept. We could intuitively find the proximity of predictions and the observations by the error rate. When the error rate is smaller, the forecasting accuracy is higher.

$$Error\ Rate = \frac{Observation - Prediction}{Observation}. \tag{9}$$

## 3. Results

Due to the transformation of the dataset from the time series to a spatial-temporal series, as well as the deficiency of precipitation around the Humber River, we use the dataset from five different stations near each other. According to the $MSE$, the training error and validation error by 12 h-ahead for the LSTM model, ConvLSTM model, CNN-LSTM model, and STA-LSTM model are plotted in Figure 7.

By building a longer forecasting time, residents in flood risk areas will have adequate time to evacuate, ensuring the highest degree of safety. Preceding this, the 24 h-ahead forecasting model is run to determine the training and validation error. The training and validation error for the 24 h-ahead LSTM model, ConvLSTM model, CNN-LSTM model, and STA-LSTM model are given in Figure 8.

Comparing the variation trend training loss and validation loss, we can judge the learning state of the model and the problems of a dataset. Then, we change the quantity and size of layers to improve the performance of the models. Additionally, we list the $MSE$ and $MAE$ for each hour ahead.

Tables 1–3 show the results of $MSE$, $MAE$, and the error rate, respectively. When the forecasting time increases, the $MSE$, $MAE$, and error rate also increases. However, the STA-LSTM model has the better performance because the $MSE$, $MAE$, and error rate of 24 h-ahead forecasting are the lowest, as shown with the red value.

We find that the $MSE$, $MAE$, and ER are not sufficient to prove model performance, so we use the Fisher test (F-test) to confirm that the performance of the STA-LSTM model is better than the other three. For the aim, the F-test applies the *F-ratio* ($F_{ratio}$) criterion. An F-test is a statistical analysis test, built within a certain confidence interval, to help distinguish the accuracy of the model prediction. The test takes into account the experimental and model uncertainties to evaluate the performance of the models. To perform the F-test analysis, a significance level and $F_{ratio}$ value must be computed. The status of the hypotheses can either be accepted or denied based on the $F_{ratio}$ value that is defined as

$$F_{ratio} = \frac{MSR}{MSE}. \tag{10}$$

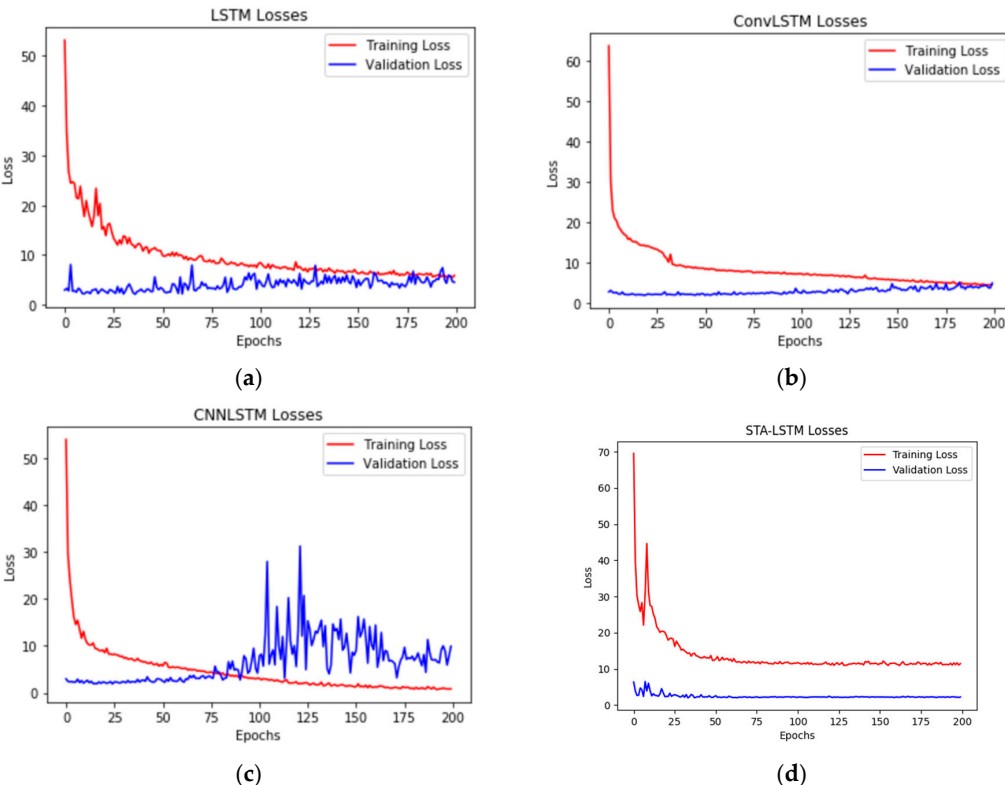

**Figure 7.** Training error and validation error by 12 h-ahead for the LSTM model (**a**), ConvLSTM model (**b**), CNN-LSTM model (**c**), and STA-LSTM model (**d**).

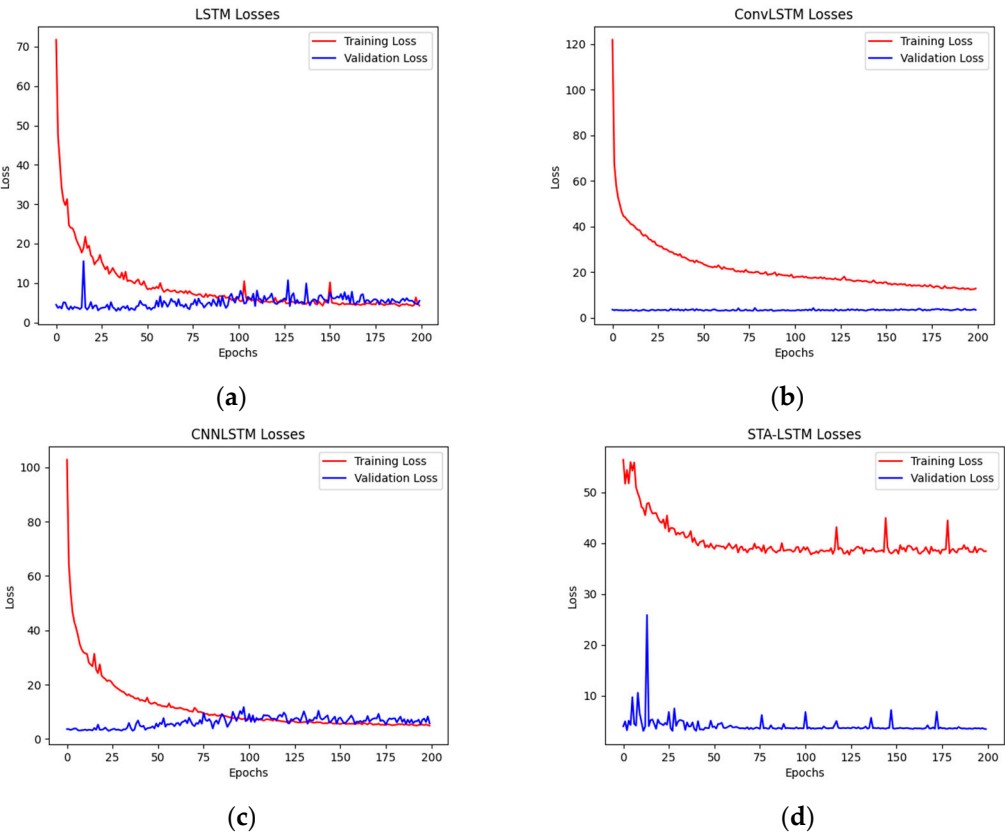

**Figure 8.** Training error and validation error by 24 h-ahead for the LSTM model (**a**), ConvLSTM model (**b**), CNN-LSTM model (**c**), and STA-LSTM model (**d**).

**Table 1.** Mean Square Error of the proposed models' performance.

| Forecasting Time | LSTM | Conv-LSTM | CNN-LSTM | STA-LSTM |
|---|---|---|---|---|
| 1 | 4.99 | 5.35 | 2.26 | 3.47 |
| 2 | 5.47 | 6.15 | 3.47 | 4.98 |
| 3 | 7.67 | 7.66 | 5.25 | 6.44 |
| 4 | 9.71 | 9.14 | 7.07 | 7.57 |
| 5 | 11.43 | 10.79 | 9.02 | 8.49 |
| 6 | 12.83 | 12.02 | 10.39 | 9.37 |
| 7 | 14.19 | 13.58 | 11.86 | 10.23 |
| 8 | 16.57 | 15.09 | 13.49 | 11.04 |
| 9 | 18.12 | 16.91 | 15.23 | 11.85 |
| 10 | 20.27 | 18.96 | 17.38 | 12.61 |
| 11 | 22.04 | 21.17 | 19.15 | 13.31 |
| 12 | 24.03 | 24.09 | 21.92 | 14.07 |
| 13 | 40.61 | 33.90 | 48.60 | 29.26 |
| 14 | 43.61 | 37.95 | 52.74 | 32.35 |
| 15 | 46.93 | 41.92 | 56.45 | 35.69 |
| 16 | 50.23 | 46.46 | 60.78 | 39.19 |
| 17 | 53.43 | 50.86 | 64.55 | 42.76 |
| 18 | 56.26 | 55.22 | 68.09 | 46.30 |
| 19 | 59.06 | 59.68 | 71.44 | 49.72 |
| 20 | 61.61 | 63.87 | 74.93 | 52.99 |
| 21 | 64.26 | 67.49 | 78.11 | 56.06 |
| 22 | 66.78 | 70.79 | 80.69 | 58.89 |
| 23 | 69.47 | 73.83 | 83.33 | 61.49 |
| 24 | 71.89 | 76.26 | 85.43 | 63.92 |

**Table 2.** Mean Absolute Error of the proposed models' performance.

| Forecasting Time | LSTM | Conv-LSTM | CNN-LSTM | STA-LSTM |
|---|---|---|---|---|
| 1 | 0.84 | 0.77 | 0.63 | 0.53 |
| 2 | 0.83 | 0.86 | 0.71 | 0.57 |
| 3 | 0.92 | 0.94 | 0.73 | 0.62 |
| 4 | 1.02 | 0.94 | 0.83 | 0.69 |
| 5 | 1.09 | 1.01 | 0.89 | 0.75 |
| 6 | 1.16 | 1.05 | 0.95 | 0.81 |
| 7 | 1.23 | 1.11 | 1.04 | 0.87 |
| 8 | 1.35 | 1.18 | 1.11 | 0.93 |
| 9 | 1.43 | 1.25 | 1.21 | 0.99 |
| 10 | 1.51 | 1.33 | 1.28 | 1.05 |
| 11 | 1.59 | 1.41 | 1.35 | 1.11 |
| 12 | 1.68 | 1.52 | 1.45 | 1.16 |
| 13 | 2.28 | 1.99 | 2.49 | 1.91 |
| 14 | 2.38 | 2.09 | 2.59 | 2.00 |
| 15 | 2.49 | 2.19 | 2.67 | 2.10 |
| 16 | 2.58 | 2.30 | 2.79 | 2.20 |
| 17 | 2.68 | 2.40 | 2.89 | 2.30 |
| 18 | 2.75 | 2.50 | 2.98 | 2.39 |
| 19 | 2.84 | 2.59 | 3.09 | 2.48 |
| 20 | 2.92 | 2.71 | 3.19 | 2.57 |
| 21 | 3.01 | 2.79 | 3.29 | 2.66 |
| 22 | 3.08 | 2.88 | 3.36 | 2.73 |
| 23 | 3.17 | 2.94 | 3.45 | 2.81 |
| 24 | 3.25 | 3.02 | 3.52 | 2.88 |

**Table 3.** Error Rate of the proposed models' performance.

| Forecasting Time (Hours-Ahead) | Module | Error Rate |
|:---:|:---:|:---:|
| 6 | LSTM | 6.43% |
| 6 | ConvLSTM | 9.69% |
| 6 | CNN-LSTM | 8.62% |
| 6 | SAT-LSTM | 3.96% |
| 12 | LSTM | 6.98% |
| 12 | ConvLSTM | 10.23% |
| 12 | CNN-LSTM | 9.87% |
| 12 | SAT-LSTM | 3.98% |
| 24 | LSTM | 8.08% |
| 24 | ConvLSTM | 11.31% |
| 24 | CNN-LSTM | 10.95% |
| 24 | SAT-LSTM | 6.31% |

A higher $F_{ratio}$ value indicates a more suitable model [57]. The $MSE$ is given in Table 1, and the mean square regression ($MSR$) is defined as

$$MSR = \frac{SSR}{k}, \tag{11}$$

$$SSR = \sum_{i=1}^{n} \left( (D_i)_p - \overline{D_o} \right)^2, \tag{12}$$

where $(D_i)_p$ is the $i$-th prediction value, $(D_i)_o$ is the $i$-th observation, $\overline{D_o}$ is the mean of $(D_i)_o$, $n$ is the number of data samples, and $k$ is the number of input variables.

Then we get the $F_{ratio}$ as given in Table 4.

**Table 4.** The F-test for the proposed models.

| Forecasting Time (Hours-Ahead) | Module | $F_{ratio}$ | Status of Hypothesis |
|:---:|:---:|:---:|:---:|
| 6 | LSTM | 1.53 | Accept |
| 6 | ConvLSTM | 1.84 | Accept |
| 6 | CNN-LSTM | 2.51 | Accept |
| 6 | SAT-LSTM | 2.91 | Accept |
| 12 | LSTM | 0.77 | Accept |
| 12 | ConvLSTM | 0.90 | Accept |
| 12 | CNN-LSTM | 1.35 | Accept |
| 12 | SAT-LSTM | 1.62 | Accept |
| 24 | LSTM | 0.53 | Accept |
| 24 | ConvLSTM | 0.81 | Accept |
| 24 | CNN-LSTM | 1.02 | Accept |
| 24 | SAT-LSTM | 1.15 | Accept |

Although all the proposed models are accepted by the $F_{ratio}$, the $F_{ratio}$ of the STA-LSTM model is better than the other three.

Furthermore, we use the uncertainty and reliability to control the accuracy level of the under-study models in a certain domain [58]. An uncertainty analysis is performed to restrict the true value of an experimental outcome. The uncertainty interval is given as:

$$UI = \overline{X} + Z\frac{S}{\sqrt{n}}, \tag{13}$$

where $\overline{X}$ is the sample average, $Z$ is 1.960, and $S$ is the sample standard deviation. This can be completed using an uncertainty interval of U95, meaning 95 out of 100 experiments completed will lie within the given interval [58]. The equation is:

$$U95 = \left(\frac{1.96}{n}\right)\sqrt{\sum_{i=1}^{n}\left((D_i)_o - \overline{(D)_o}\right)^2 + \sum_{i=1}^{n}\left((D_i)_o - (D_i)_p\right)^2}. \qquad (14)$$

In the four flood forecasting proposed models, the STA-LSTM model had the lowest uncertainty value ($U95 = 0.2051$) when the forecasting time is 24 h-ahead, while the LSTM model ($U95 = 0.2105$) is the highest value of uncertainty in Table 5. Therefore, in terms of U95, the STA-LSTM model outperformed the other three hybrid LSTM models. Then, a reliability analysis was conducted to statistically determine the overall model consistency. The two equations used in the analysis are as follows:

$$RAE_i = \left| \frac{(D_i)_o - (D_i)_p}{(D_x^*)_{i(o)}} \right|, \qquad (15)$$

$$Reliability = \left(\frac{100\%}{n}\right)\sum_{i=1}^{n} k_i, \qquad (16)$$

**Table 5.** The U95 for the proposed models.

| Forecasting Time (Hours-Ahead) | Module | U95 |
|---|---|---|
| 6 | LSTM | 0.2001 |
| 6 | ConvLSTM | 0.1983 |
| 6 | CNN-LSTM | 0.1961 |
| 6 | SAT-LSTM | 0.1969 |
| 12 | LSTM | 0.2092 |
| 12 | ConvLSTM | 0.2073 |
| 12 | CNN-LSTM | 0.2045 |
| 12 | SAT-LSTM | 0.2042 |
| 24 | LSTM | 0.2105 |
| 24 | ConvLSTM | 0.2085 |
| 24 | CNN-LSTM | 0.2058 |
| 24 | SAT-LSTM | 0.2015 |

If the relative average error (RAE) is less than the threshold value of an adequate water quality parameter, the $k_i = 1$, meaning the ki is the amount that the RAE is less than or equal to the water quality parameter [59]. The optimum value is 0.2, according to the Chinese Standards.

From Table 6, it is conspicuous that the STA-LSTM model, with $Reliability = 20.48\%$ and 21.05%, was the most reliable model of the four proposed models when the forecasting times are 12 h-ahead and 24 h-ahead.

**Table 6.** The reliability for the proposed models.

| Forecasting Time (Hours-Ahead) | Module | Reliability |
|---|---|---|
| 6 | LSTM | 17.55% |
| 6 | ConvLSTM | 21.75% |
| 6 | CNN-LSTM | 22.40% |
| 6 | SAT-LSTM | 22.34% |
| 12 | LSTM | 14.30% |
| 12 | ConvLSTM | 22.02% |
| 12 | CNN-LSTM | 22.09% |
| 12 | SAT-LSTM | 20.48% |
| 24 | LSTM | 13.95% |
| 24 | ConvLSTM | 22.51% |
| 24 | CNN-LSTM | 22.56% |
| 24 | SAT-LSTM | 21.05% |

## 4. Discussion

The loss plots show that the best validation performance happened, roughly, in epoch 200 for LSTM models and ConvLSTM models. The best validation performance happened, roughly, in epoch 75 for the CNN-LSTM model with the 12 h-ahead forecasting, and the best validation performance happened, roughly, in epoch 100 for the LSTM model with the 24 h-ahead forecasting. Moreover, from Figures 7 and 8, the validation loss is less than the training loss because the regularization is applied at the training but not during validation. The second reason is that the training losses were measured during each epoch, while validation losses were measured after each epoch.

### 4.1. The 12 h-Ahead Forecasting

As seen in the training and validation error plot, shown as Figure 7a, for the 12 h-ahead measurement of the LSTM model, the training loss decreased, and the validation loss steadily increased; therefore, the training loss would be fitting with the validation loss and keep stead.

As observed in the plot of training error and the validation error, shown as Figure 7b, by 12 h-ahead for the ConvLSTM model, the training loss decreases, and the validation loss increases slowly; therefore, the training loss would be fitting with the validation loss and keep stead.

As shown in the plot of the training error and validation error of the 12 h ahead measurement for the CNN-LSTM model, seen in Figure 7c, the training loss decreases, and the validation loss increases slowly before fitting. The training loss would be fitting with the validation loss at epoch 75, and the training loss still decreases gradually. However, the validation loss has violently oscillated, so the validation data is scarce and not very representative of the training data [60].

The plot of the training error and validation error of the 12 h-ahead measurement for the STA-LSTM model in Figure 7d shows that the validation loss is much better than the training loss, thus reflecting that the validation dataset is easier to predict than the training dataset.

Moreover, Figure 7 shows the performance of the 12 h-ahead flood forecasting models. The results were created using the training validation data to determine the losses, based on a statistical analysis, using the LSTM models. The validation and training losses can be analyzed on a graph of loss and epoch number. Epoch numbers were chosen to create an optimal fit in the data, which neither under nor overfits. It can be observed that the LSTM and ConvLSTM models had the highest performance, as indicated by the good fit relationship at approximately 200 epochs. The STA-LSTM and CNNLSTM model proved to be less than optimal, as neither displayed a good fit relationship.

### 4.2. The 24 h-Ahead Forecasting

Building models for the long-term forecasting of floods will provide more time for people to evacuate in the case of flood disasters.

As observed in the training and validation error plot, shown as Figure 8a, for the 24 h-ahead measurement of the LSTM model, the training loss decreased, and the validation loss steadily increased. Then, the training loss would be fitting with validation loss at the epoch 100, and the training loss still decreases gradually and keep stead.

As observed in the plot of training error and validation error, shown as Figure 8b, by the 24 h-ahead measurement for the ConvLSTM model, the training loss decreases, while the validation loss increases slowly and keeps stead. The training loss would be fitting with validation loss at about epoch 300.

As shown in the plot of the training error and validation error of the 24 h-ahead for the CNN-LSTM model, seen in Figure 8c, the training loss decreases, the validation loss increases slowly before fitting, and the training loss would be fitting with the validation loss at the epoch 100. After fitting, the validation loss has the small amplitude oscillation around the training loss.

The plot of the training error and validation error by the 24 h-ahead measurement for the STA-LSTM model in Figure 8d shows that the validation loss is much better than the training loss, reflecting that the validation dataset is easier to predict than the training dataset, which is the same as the 12 h-ahead forecasting. When the forecasting time increases to 48-h-ahead, the model produces a training loss of 95 and validation of 10, which shows the instability of the model and that it needs improvements.

Moreover, Figure 8 shows the performance of the 24 h-ahead flood forecasting models, with results synthesized using the same procedure as the 12 h-ahead models. It can be observed that the LSTM and CNNLSTM models are most optimal, as they reach a good fit relationship at approximately 200 epochs. The ConvLSTM model can be seen as slightly underfitting; however, it is not as underfitting as the STA-LSTM model, which was severely underfitted. This is an indication that that the model is unsuitable to model the training data. In our opinion, when the model of loss plot happened to be underfitting, we could add epochs, shuffle parts, and increase the hidden node to improve the STA-LSTM model to fitting.

Furthermore, the results indicate four proposed models. When regularization is applied to the validation, the results indicate that the four models could forecast the discharge of the Humber River with a *MAE* of less than 0.45 m$^3$/s, as indicated in Tables 1 and 2.

Table 1 presents the *MSE* results of LSTM, Conv-LSTM, CNN-LSTM, and STA-LSTM models for the 24 h-ahead forecasting. The CNN-LSTM model produced the best results (*MSE* = 2.26) for the furcating time of 1 h-ahead. The STA-LSTM model produced the best results (*MSE* = 63.92) for the forecasting time of 24 h-ahead. From all experimental results, we observed that, as the forecasting time increased, the value of *MSE* also increased. Overall results show that STA-LSTM produced the best results and outperformed all other models. The results indicate that the CNN-LSTM model achieved poor results (*MSE* = 85.43) for the forecasting time of 24 h.

Table 2 presents the *MAE* results of LSTM, Conv-LSTM, CNN-LSTM, and STA-LSTM models for the 24 h-ahead forecasting. The STA-LSTM model produced the best results (*MAE* = 0.53) for the forecasting time of 1 h-ahead and the best results (*MAE* = 2.88) for the forecasting time of 24 h-ahead. From all experimental results, we observed that as the forecasting time increased the value of *MAE* also increased. The overall results show that STA-LSTM produced the best results and outperformed all other models. The results indicate that the CNN-LSTM model achieved poor results (*MAE* = 3.52) for the forecasting time of 24 h.

To summarize, the CNN-LSTM has good performance when the forecasting time is less than four hours ahead, due to the *MSE* of 1-h-ahead being 2.26 and the *MSE* of 2-h-ahead being 3.47, which is less than the STA-LSTM model. Therefore, we can further debug the CNN-LSTM model parameters. Moreover, this research shows that the forecasting performance of hourly discharge can be boosted using the STA-LSTM model due to the error rate prediction being lowest at about 6.31%, as provided in Table 3.

The hybrid LSTM models can be compared with the results of previous studies to assess the superiority of flood predictions. Previously, different artificial neural network models could be applied for short-term flood forecasting, including the M5 model tree, ELM, and ANN. The *MSE* and *MAE* are used to perform a comparison of models. As stated in Tiwari et al., for a forecasting time of 1 h the ANN produced an *MAE* of 26.26, ELM obtained 0.292, and the M5 model achieved 0.291. The *MAE* for the hybrid LSTM models was obtained to be 0.84 for LSTM, 0.77 for ConvLSTM, 0.63 for CNNLSTM, and 0.53 for STA-LSTM [61]. The experimental results of our proposed models show that the *MAE* is higher compared to the previously reported values determined by Tiwari et al. using ELM and the M5 model tree. Both the hybrid LSTM models, ELM and M5 model tree are very suitable for hydrological modeling, however, differ in structure. The architecture of the ELM model is similar to the ANN model, which contains an input layer, output layer, and at least one hidden gate. The M5 model tree is a linear regression model which is mostly used for numerical predictions of variables. The mean average error results show that the

ELM and M5 models are suitable for hydrological analysis but were not used for flood forecasting problems in this study, as our goal was to utilize the new STA-LSTM model. In terms of percentage, there is quite a large error between the hybrid LSTM models, even with the lowest *MAE* of the STA-LSTM model. The ANN model conducted by Tiwari et al., however, produces a very large error, which is an indication that the hybrid LSTM models produce a similar accuracy to other model options.

In addition, a comparison of proposed hybrid models can be performed using LSTM models from the literature to determine the accuracy and precision of models tested by Ding et al. The *MAE* and *MSE* of our models can be seen to be much lower than that of CNN, GCN, LSTM, and STA-LSTM in Ding et al. The CNN model produced a *MAE* of 38.29, GCN had a *MAE* of 38.15, LSTM was with 38.31, and STA-LSTM was with 37.49. As the models have a similar structure, the accuracy can be compared to determine the superiority. Our proposed models produced promising results that proved to have the highest accuracy when compared to the models of similar structure in Ding et al. Our proposed model analysis includes Hybrid models that achieved the state-of-the-art results and outperformed the previously reported results in the literature by Ding et al. All in all, we think the spatio-temporal series dataset could improve the performance of the hybrid LSTM models.

Comparing the four proposed models, we find that the hybrid LSTM model has had the best performance when the attention was applied to the hybrid LSTM model. We compare the prediction result of validation part with observations in Figure 9, the STA-LSTM model more robustly forecasts hourly discharge than other models when the forecasting time increases. This is due to the forecasting time of most published papers being within 12 h [43]. When the forecasting time increases to 12 h, the STA-LSTM still has robust performance. From the Figure 9d, we find the contact ratio of predication and observation is higher in the STA-LSTM plot. The LSTM plot has the lowest contact ratio than other three models in the Figure 9a. Then the ConvLSTM plot is similar to CNNLSTM plot in the Figure 9b,c, respectively. The training model is workable for flood forecasting. However, the forecasting currency needs to be improved in the rainy season, from June to August, and we find that more features are needed as input to improve the performance of STA-LSTM model when the forecasting time increases to 24 h. Moreover, comparing the results with the literature on flood forecasting problems, our proposed models produced promising results [62]. Our proposed models include Hybrid models that achieved the state-of-the-art results and outperformed the previously reported results in the literature.

The overall results indicate that LSTM-based models are more suitable for sequence-based data to perform forecasting analysis, and they present highly reliable results for flood forecasting problems. Overall, all variations of LSTM models produced reliable performances in terms of error rate. The STA-LSTM model produced more reliable and efficient results, for flood forecasting at all hours, and outperformed other models due to the U95 value and reliability values.

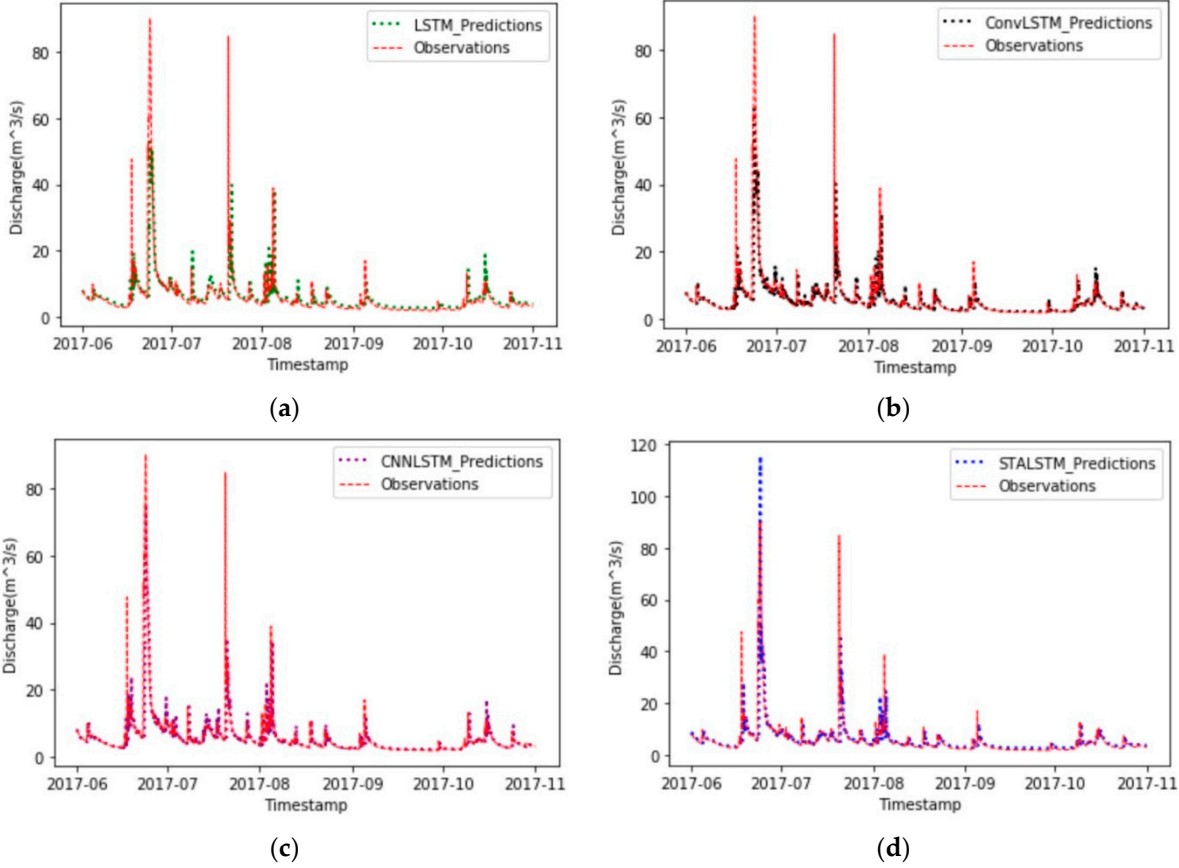

**Figure 9.** Comparing the predictions and the observations. (**a**) LSTM model, (**b**) ConvLSTM model, (**c**) CNN-LSTM model, (**d**) STA-LSTM model.

## 5. Conclusions

Forecast accuracy concerning the magnitude and the timing of the flood water levels diminishes significantly with forecast time, which is a critical aspect of an early warning system. Only models that can accurately predict flood water levels with sufficient warning time to allow safe evacuation can be useful tools. Our work has advanced flood forecasting accuracy, using spatio-temporal tools and deep learning algorithms to utilize the newly established real-time river monitoring network.

The new models presented here will be helpful for governments, insurance companies, local authorities, and first respondents to manage major flood events effectively. This study focused on summer thunderstorm events that are the dominant process for urban areas such as the city of Toronto. The STA-LSTM model has better performance for the summer thunderstorm events, as shown by the forecasting lowest error rate at about 3.98% for a 12-h-ahead prediction.

Almost all floods from extreme climate, such as torrential rain and global warming (snowmelt, ice jam, etc.), would require building the spatio-temporal relationship between the flow, the air temperature, the precipitation, and the snow depth. Therefore, for future work, we will face complex dataset pre-processing, such as normalization, due to the different units. We plan to test and compare the STA-LSTM model and the Spatio-temporal Attention Gated Recurrent Unit (STA-GRU) model, as well as the Generative Adversarial Networks Long Short-term Memory (GAN-LSTM) model. Including GAN models might help accuracy as the spatio-temporal dataset sizes increase. We will add more features, such as snow depth surveys, air temperature, and precipitation, as model inputs to improve the spring snowmelt floods' accuracy, which are the dominant process in rural watersheds in Canada.

**Author Contributions:** Conceptualization: Y.Z.; methodology: Y.Z. and Z.G.; software: J.V.G.T., Y.Z. and Z.G.; validation: Y.Z.; formal analysis: Y.Z.; investigation: Y.Z.; resources: Y.Z. and B.G.; data curation: Y.Z.; writing—original draft preparation: Y.Z., B.G., J.V.G.T. and S.X.Y.; writing—review and editing: B.G., J.V.G.T. and S.X.Y.; supervision: B.G. and S.X.Y.; project administration: B.G., S.X.Y. and J.V.G.T.; funding acquisition: B.G. and J.V.G.T. All authors have read and agreed to the published version of the manuscript.

**Funding:** This research was funded by the Natural Sciences and Engineering Research Council of Canada (NSERC) Alliance Grant #401643.

**Data Availability Statement:** Datasets are available upon request from the corresponding author.

**Conflicts of Interest:** The authors declare no conflict of interest.

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
