# Peer review of "The Discharge Forecasting of Multiple Monitoring Station for Humber River by Hybrid LSTM Models"

_water, doi:10.3390/w14111794_

Round 1

Reviewer 1 Report

The paper presents a study aimed to improve flood forecasting accuracy, using real-time data, registered by monitoring network. In particular, the watershed of the Humber River is considered as a study case.

This is an interesting and actual topic, which fits well with the scope of the journal. However, there are a few issues that must be fixed, before considering the paper for publication.

I have found some typos, though there may be others. So, I suggest the paper be carefully double checked.

All symbols used in the paper must be explained. For example, many of the symbols in equations (1) to (6) are not explained at all.

The presentation of the models is very poor. Figures (4) to (7) are definitely not enough to understand how the models work! The figures must be explained and commented. The models must be better presented.

Page 8. I don’t understand the reason for points 5, 6 and 7. These do not belong to the list of considered models. Moreover, at line 259, “Set up the error rate concept.”: is this a title?

Lines 273-274 and 337-338. Please, rewrite these sentences.

Presentation and discussion of the results is also very poor. For example, figures (8) and (9) must be better explained and discussed.

Author Response

COMMENT #1: The paper presents a study aimed to improve flood forecasting accuracy, using real-time data, registered by monitoring network. In particular, the watershed of the Humber River is considered as a study case. This is an interesting and actual topic, which fits well with the scope of the journal. However, there are a few issues that must be fixed, before considering the paper for publication.

RESPONSE: We appreciate the time and effort that you have dedicated to providing your insightful comments on our paper and we have incorporated changes to reflect your suggestions.  These changes are highlighted within the manuscript using track changes. Below is a point-by-point response your comments.

COMMENT #2: I have found some typos, though there may be others. So, I suggest the paper be carefully double checked. All symbols used in the paper must be explained. For example, many of the symbols in equations (1) to (6) are not explained at all.

RESPONSE: Thanks for your suggestion. The mathematical equations of each layer of the LSTM model are given in equations (1) to (6).  In the equations, i_t represents the input gate, o_t represents the output gate, f_t represents the forget gate, C ̃_t represents the memory cell,  h_t represents a hidden state, h_(t-1) and C_(t-1) represents the inputs of previous timestamps, x_t denotes the current timestamp, σ symbolizes the sigmoid function,  b_x represents the bias of respective gates, and  W_x represents the weights of respective gates.” (Please see Lines 249-255 on Page 7.)

Reference: Tabrizi SE, Xiao K, Thé JV, Saad M, Farghaly H, Yang SX, Gharabaghi B. Hourly road pavement surface temperature forecasting using deep learning models. Journal of Hydrology. 2021 Dec 1;603:126877.

COMMENT #3: The presentation of the models is very poor. Figures (4) to (7) are definitely not enough to understand how the models work! The figures must be explained and commented. The models must be better presented.

RESPONSE: Thank you for the comment. We revised the text to better explain Figures (4) to (7) as follows: “The structure of the LSTM model is presented in Figure 4, which is comprised of LSTM layers.  In the structure, input sequences are provided to the input layer followed by two LSTM layers.  The dropout layer is added to prevent the model from overfitting then two LSTM layers are added followed by a Flatten layer.  Three dense layers are added followed by one dropout and three dense layers which are used to change the dimensions of the vectors.  Finally, the last output layer returns the output sequences.”  Figures 4 to 7 are adopted from published work, and we have included citations and the references to the original papers and the code for the readers.  (Please see Lines ?-? on Page ?.)

COMMENT #4: Page 8. I don’t understand the reason for points 5, 6 and 7. These do not belong to the list of considered models. Moreover, at line 259, “Set up the error rate concept.”: is this a title?

RESPONSE: Thank you for your comment.  Points 5, 6 and 7 discuss the reasoning and thinking behind our machine learning model performance tests and error minimization for model training and testing. We have revised the text to improve clarity. We renew the subsection for them that the points 1, 2, 3, 4 in the subsection “model” and the point 5, 6, 7 in the subsection “Validation method”.  Moreover, the line 259, “Set up the error rate concept.”, is not a title, and We update the line 259 as “The Error Rate is used as the third evaluation measure concept.  We could intuitively find the proximity of predictions and the observations by the error rate.  When the error rate is smaller, the forecasting accuracy is higher.”

COMMENT #5: Lines 273-274 and 337-338. Please, rewrite these sentences.

RESPONSE: Thank you for the suggestion.  We rewrote these sentences to make the expressed ideas clearer to readers. The lines 273-274 revised text reads: “By building a longer forecasting time, residents in the flood inundation area will have adequate time to equate ensuring the highest degree of safety.  Proceeding this, the 24 hour-ahead forecasting model is run to determine training and validation error.” We update to: “By building a longer forecasting time, residents in flood risk areas will have ade-quate time to evacuate ensuring the highest degree of safety. Proceeding this, the 24 hour-ahead forecasting model is run to determine the training and validation error. The training and validation error for the 24 hour-ahead LSTM model, ConvLSTM model, CNN-LSTM model and STA-LSTM model are given in Figure 9.” (Please see Lines ?-? on Page ?.)

And the lines 337-338 revised text reads: “The results indicate four proposed models. Therefore, applying regularization to the validation. The results indicate that the four proposed models could forecast the discharge of the Humber River with a MAE of less than 0.45 m^3/s from the Tables 1 and 2.” We update to: “The results indicate four proposed models. When regularization is applied to the validation, the results indicate that the four models could forecast the discharge of the Humber River with a MAE of less than 0.45 m3/s indicated in Tables 1 and 2. (Please see Lines ?-? on Page ?.)

COMMENT #6: Presentation and discussion of the results is also very poor. For example, figures (8) and (9) must be better explained and discussed.

RESPONSE: Thank you for highlighting this issue. Lines 391-398 of the revised manuscript, under the subsection “4.1 The 12 Hour-ahead Forecasting” we have revised the text as follows: “Moreover, Figure 8 shows the performance of the 12 hours-ahead flood forecasting models. The results were created using the training validation data to determine the losses based on a statistical analysis using the LSTM models.  The validation and training losses can be analyzed on a graph of loss and epoch number.  Epoch numbers were chosen to create an optimal fit in the data which neither under nor overfits. It can be observed that the LSTM and ConvLSTM models had the highest performance as indicated by the good fit relationship at approximately 200 epochs.  The STA-LSTM and CNNLSTM model proved to be less then optimal as neither displayed a good fit relationship.”

Lines 422-433 of the revised manuscript, in the subsection “4.2 The 24 Hour-ahead Forecasting”: “Moreover, figure 9 shows the performance of the 24 hours-ahead flood forecasting models, with results synthesized using the same procedure as the 12 hours-ahead models.  It can be observed that the LSTM and CNNLSTM models are most optimal as they reach a good fit relationship at approximately 200 epochs.  The ConvLSTM model can be seen as slightly underfitting however, not as much as the STA-LSTM model which was severely underfitted.  This is an indication that that the model is unsuitable to model the training data. In our opinion, when the model of loss plot happened the underfitting, we could add epochs and shuffle part, and increase the hidden node to improve the STA-LSTM model to fitting.”

Reference: Tabrizi SE, Xiao K, Thé JV, Saad M, Farghaly H, Yang SX, Gharabaghi B. Hourly road pavement surface temperature forecasting using deep learning models. Journal of Hydrology. 2021 Dec 1;603:126877.)

Reviewer 2 Report

The use of pattern recognition approaches for flow forecasting has a long history.  However, the focus of the authors in their literature analysis has been on recent literature where simplifications of many past studies are reported; this results in subtle points previously reported being neglected in this study.  For example, difficulties in predicting changes in flow regime while being able to predict dry weather or flood flow satisfactorily, are not discussed in this study.  Furthermore, the concept of multi-models with variable belief structures has not been raised as an approach to obtain reliable predictions.

Some more specific errors in the manuscript include:

1 - Line 83.  The statement is incorrect.  While the citation may be correct, there have been SWMM models applied to New York and other large cities (see, for example, Sun et al., 2014)

2 - Lines 100 to 117.  The focus here is the spatio-temporal variability of data relevant to flood forecasting.  Even though rainfall is the dominant source of this variabiltiy, there is an absence of references to rainfall processes.  Knowledge of the data and the source of its variability is extremely important.  Equally important is the uncertainty associated with the data - it is impossible to measure rainfall with absolute accuracy.

3 - In any flood forecasting system, warning time is important.  This is usually a function of the catchment response time.  While the Authors proposed approach does not need the catchment response time, there is no mention of warning times - is the time from rainfall occurrence to flood event 1 hour or 12 hours?  This is not discussed.

Author Response

COMMENT #1: In their literature analysis has been on recent literature where simplifications of many past studies are reported; this results in subtle points previously reported being neglected in this study.  For example, difficulties in predicting changes in flow regime while being able to predict dry weather or flood flow satisfactorily, are not discussed in this study.  Furthermore, the concept of multi-models with variable belief structures has not been raised as an approach to obtain reliable predictions.

RESPONSE: We appreciate the time and effort that you dedicated to providing this valuable feedback on our manuscript.  We are grateful for your insightful comments on our paper and have incorporated changes to reflect your suggestions.  These changes are highlighted within the manuscript using track changes.  Below is a point-by-point response your comments.

Some more specific errors in the manuscript include:

COMMENT #2: Line 83.  The statement is incorrect.  While the citation may be correct, there have been SWMM models applied to New York and other large cities (see, for example, Sun et al., 2014)

RESPONSE: Thank you for bringing attention to this reference. We have revised the text in Line 83 as follows: “Physical hydrological models were developed to simulate flooding events in cities such as the Environmental Protection Agency Storm Water Management Model (EPA-SWMM). Still, these models are not available for older and larger cities [17].  We update the line 83 as follows: “Still, these models are not available for older and larger megacities due to the SWMM relay to the high precision mapping and a simulation of the underground drainage system [17] .”

Reference: Sun, Ning, Myrna Hall, Bongghi Hong, and LianJun Zhang. "Impact of SWMM catchment discretization: Case study in Syracuse, New York." Journal of Hydrologic Engineering 19, no. 1 (2014): 223-234.)

COMMENT #3: Lines 100 to 117.  The focus here is the spatio-temporal variability of data relevant to flood forecasting.  Even though rainfall is the dominant source of this variabiltiy, there is an absence of references to rainfall processes.  Knowledge of the data and the source of its variability is extremely important.  Equally important is the uncertainty associated with the data - it is impossible to measure rainfall with absolute accuracy.

RESPONSE: Thank you for this insightful comment. The network of real-time tipping-bucket rainfall monitoring in the Humber River watershed is sparse and does not accurately capture the spatial variability of the intense localized summer thunderstorms.  Weather Radar data is available to combine with rain gauge data, but it requires massive computational time/efforts (we have published a few papers on this topic – reference list provided below).  Moreover, the raw rainfall data must be first pre-processed and accumulated for the watershed over time and space before it can provide meaningful input to the machine learning model.  Our preliminary attempts to take advantage of the raw real-time rainfall monitoring data as additional input for the training of the machine learning models did not yield any improvement in the accuracy of the flood forecasts.  Therefore, to keep the flood forecasting system simple and practical, yet fairly accurate, we decided not to include rainfall monitoring data as part of the scope of this manuscript.  New paragraph has been added to the revised manuscript discuss this important point (please see page 5 Lines 225 – 231).

COMMENT #4: In any flood forecasting system, warning time is important.  This is usually a function of the catchment response time.  While the Authors proposed approach does not need the catchment response time, there is no mention of warning times - is the time from rainfall occurrence to flood event 1 hour or 12 hours?  This is not discussed.

RESPONSE: Thank you very much for your perceptive comment.  Flood warnings are distinct from forecasts, since they are issued when an event is imminent or already occurring.  Due to the level of urbanization and size of the Humber River watershed, the catchment response time typically ranges from 5 to 10 hours depending on the rainfall storm event type and duration (e.g. short-burst but intense summer thunder storms versus longer duration rainfall combined with snowmelt events in the spring).  The Flood warnings for the Humber River watershed must be issued to a range of users and for various purposes.  These purposes may include readying operational teams and emergency personnel; warning the public of the timing and location of the event; and in extreme cases, to enable preparation for undertaking evacuation and emergency procedures.  Therefore, we train/test the model for 12-hours-ahead and 24-hours-ahead forecast scenarios and evaluated model accuracy.  New paragraph has been added to the revised manuscript to discuss this important point (please see Lines 253 – 262 of the revised manuscript).

Reviewer 3 Report

Authors made formidable efforts to predict the discharge of multiple monitoring station for Humber River by Hybrid LSTM Models. The paper was overall clearly written and organized. However, in the case of comparisons, authors can improve their results representation:

(1) Why did authors use LSTM in comparison with other robust soft computing models such as M5MT, MARS, EPR, GEP, ELM, and SVM?? This issue needs major clarifications.

(2) Results of hybrid LSTM models can be compared with literatures

(3) Uncertainty analysis and F-test of LSTM results requires to be added into results section. This analysis can be found in literature: Water Resources Management 34, 529–561, 2020; 

(4) Reliability analysis of LSTM results requires to be added into results section. This analysis can be found in literature: Water Resources Management 35, 3703–3720, 2021. 

Round 2

Reviewer 1 Report

Dear Authors, thank you for detailed answers on all comments and improving paper. In my opinion current version of the paper is better than previous one. I have not additional comments.

Reviewer 2 Report

The modifications and responses prepared by the Authors have addressed the previous comments.  While I disagree with the Authors on the differences between Flood Warning and Flood Forecasting, this is an argument in semantics.  Irrespective of this issue, the question posed in the comment within the original review was adequately answered in their response.

Reviewer 3 Report

I recommended acceptance of the revised paper.